# Potential of Interleukin (IL)-12 Group as Antivirals: Severe Viral Disease Prevention and Management

**DOI:** 10.3390/ijms24087350

**Published:** 2023-04-16

**Authors:** Nur Azizah A. Rahman, Vinod R. M. T. Balasubramaniam, Wei Boon Yap

**Affiliations:** 1Center for Toxicology and Health Risk Studies, Faculty of Health Sciences, Universiti Kebangsaan Malaysia, Jalan Raja Muda Abdul Aziz, Kuala Lumpur 50300, Malaysia; p99917@siswa.ukm.edu.my; 2Jeffrey Cheah School of Medicine & Health Sciences, Monash University Malaysia, Jalan Lagoon Selatan, Bandar Sunway 46150, Malaysia; vinod.balasubramaniam@monash.edu; 3Biomedical Science Program, Faculty of Health Sciences, Universiti Kebangsaan Malaysia, Jalan Raja Muda Abdul Aziz, Kuala Lumpur 50300, Malaysia

**Keywords:** cytokines, antiviral, IL-12, innate immunity, adaptive immunity

## Abstract

The interleukin (IL)-12 family consists of pro- and anti-inflammatory cytokines that are able to signal the activation of host antiviral immunity while preventing over-reactive immune reactions due to active virus replication and viral clearance. Amongst others, IL-12 and IL-23 are produced and released by innate immune cells such as monocytes and macrophages to signal the proliferation of T cells and release of effector cytokines, which subsequently activate host defence against virus infections. Interestingly, the dualities of IL-27 and -35 are evidently shown in the course of virus infections; they regulate the synthesis of cytokines and antiviral molecules, proliferation of T cells, and viral antigen presentation in order to maximize virus clearance by the host immune system. In terms of anti-inflammatory reactions, IL-27 signals the formation of regulatory T cells (Treg) which in turn secrete IL-35 to control the scale of inflammatory response that takes place during virus infections. Given the multitasking of the IL-12 family in regards to the elimination of virus infections, its potential in antiviral therapy is unequivocally important. Thus, this work aims to delve deeper into the antiviral actions of the IL-12 family and their applications in antiviral therapies.

## 1. Introduction

Disease-causing viruses can be transmitted from persons to persons or from animals to humans, leading to various disease manifestations [1,2]. Generally, there are several mechanisms that support virus entry. Enveloped viruses can enter host cells by receptor-mediated endocytosis and subsequently release their genomes into the cells to initiate genome replication [3]. Endocytosis can occur via clathrin proteins, macropinocytosis and caveolae [4]. Upon viral invasion into vulnerable host cells, the host immune response is activated, particularly the synthesis and release of pro-inflammatory cytokines such as interleukin (IL)-8 and IL-6 at significant levels [5,6]. Given the significant roles of cytokines in modulating the antiviral immunity in hosts, they are usually associated with the manifestation of viral disease symptoms and disease severity.

Unlike antimicrobial treatments for bacterial, fungal and parasitic infections, antivirals available for virus infections are very limited and their usage is also restricted to particular types of virus infections due to unique viral genetic make-ups and the architectures of antigenic viral epitopes. An antiviral treatment becomes more complex when the contagious viral agent develops resistance and abilities to inhibit cellular antiviral responses such as downregulation of type-1 interferon (IFN) signalling by degrading the signal transducer and activator of transcription (STAT)-2 [7]. As a result, efforts to seek effective antiviral drugs and therapeutic agents are required in order to better prepare the public health sector to face the occurrence of seasonal viral outbreaks and the occasional viral pandemics [8].

Given the multifactorial immune-modulatory roles of interleukins, they have been considered to be among the sought-after therapeutic molecules for treating virus infections, especially in promoting host immunity and antiviral actions [9,10]. Amongst the identified interleukins, the IL-12 family consists of cytokines with remarkable potential in inhibiting virus infections. For example, IL-12 and IL-27 have been shown to be capable of inhibiting hepatitis B virus (Hep B) and hepatitis C virus (Hep C) infections, respectively [11]. In an IL-12p40-deficient mice model, IL-12 was proven to be responsible for limiting lung inflammation and mucus production caused by human metapneumovirus (hMPV) infection [12]. The challenged mice gradually lost their lung functions and experienced an altered cytokine response following the virus infection. Considering the evidence, immune-modulation of the IL-12 family during virus infections is especially important to help restrict virus replication and regulate the host’s inflammatory response induced by virus infections. Thus, this review aims to dissect and deliberate the involvement and mechanisms of several cytokines of the IL-12 family in counteracting virus infections, and their abilities in preventing virus-induced severe disease symptoms and complications.

## 2. Activities of IL-12, IL-23, IL-27 and IL-35 during Virus Infections

The members of the IL-12 family encompass IL-12, IL-23, IL-27 and IL-35. The vast majority of the IL-12 family members are secreted by myeloid-origin cells expressing compatible receptors [13]. Despite similarities in their molecular structures, individually, these cytokines have unique actions and mechanisms in combating virus infections [11]. In this light, understanding their respective activities and functions in response to virus infections can be a great step in developing them into effective treatment agents for virus infections.

### 2.1. Pro-Inflammatory IL-12 and IL-23

Being a pro-inflammatory cytokine, IL-12 has been proven to be effective in inhibiting virus infections and ameliorating the infection symptoms when administered prophylactically and post-infection [14]. Relative to other cytokines (such as IL-18, TNF-α and IFN-α, -β, and -γ), IL-12 is synthesized more significantly in hosts to initiate the antiviral immune response as early as day-1 post-infection [15]. IL-12 is synthesized and released profoundly by innate immune cells, for instance macrophages at the early stage of infections following the binding of viral antigens to PRR (PAMP recognition receptors) such as Toll-like receptor (TLR)-3, TLR-7, and TLR-8 [16,17,18]. Upon release, IL-12 binds to the compatible receptors on T cells and regulates the expression of T-bet, the transcription factor that is responsible for CD4+ T-cell proliferation into helper T cell (Th)-1 (Figure 1). The formation of Th-1 subsequently activates the synthesis of immune effectors such as IFN-γ, tumor necrosis factor (TNF)-α, CD-107a, IL-2 and granzyme [19]. Those effectors help remove infectious virus particles and infected cells, thus preventing the systemic spread of the contagious entity in hosts. For instance, synthesis of IFN-γ by Th-1 through Janus kinase (Jak)/STAT signalling promotes the proliferation and activation of macrophages and natural killer (NK) T cells that are responsible for engulfing virus particles and clearance of virally infected cells, respectively (Figure 1) [20]. Bhardwaj et al. (1996) showed that IL-12 enhanced the proliferation of T cells at a relatively low dose in the course of a virus infection [21]. In view of its promising antiviral effects even at a relatively low dose, utilization of IL-12 in antiviral therapies can reduce the chances of patients experiencing adverse side effects that are usually seen in antiviral chemotherapy.

In terms of the cellular immune response, the activation of CD4+ T cells through the actions of IL-12 helps sustain the number of CD8+ T cells in chronic viral infections. The presence of activated CD8+ T cells is crucial in ensuring successful clearance of viral infections via cytolytic and cytotoxic mechanisms [22]. The identified cytolytic and cytotoxic mechanisms include granule protein-mediated and receptor-mediated cell death. Binding of degradative granule proteins to the cell membrane and engagement of death receptors with their ligands causes pore formation and activation of the cell death mechanism, respectively. Upon activation of cell death, membrane blebbing, nuclear DNA fragmentation and cytoplasm vacuolization take place in infected cells [23]. Altogether, it emphasizes the importance of IL-12 in antiviral immunity as it links innate and cellular immunity and promotes the production of effector antiviral biomolecules in order to expedite the elimination of viruses.

IL-23, another member of the IL-12 family, demonstrates similar regulatory properties in promoting the secretion of effector cytokines. Innate immune cells such as NK cells are actively involved in IL-23 antiviral activity. NK cells express IL-23 receptors (IL-23 Rs) on their cell surface, which initiate the expression of effector IFN-γ by NK cells upon binding to IL-23. IFN-γ is released to promote CD8+ T-cell responses such as perforin-dependent cytotoxicity in virally infected cells (Figure 2). As a result, it prevents the spread of virus infections to neighbouring host cells [24,25,26].

The roles of IL-23 in host antiviral immunity were verified using a recombinant vaccinia virus model expressing IL-23 (vv-IL-23) [27]. The recombinant virus was less virulent than the wild-type vaccinia virus in the challenged mice. The lower pathogenicity was likely attributable to the enhanced cytotoxic T cell activity mediated by IL-23. The IL-23-mediated anti-VV mechanism was also reportedly independent of IFN-γ, which nonetheless is essential to promote IL-12-mediated immunity. Surprisingly, the study also reported the involvement of IL-17 in the IL-23-regulated anti-VV response, albeit to a lesser extent. Inhibition of IL-17 using a monoclonal antibody resulted in a significant increase in vaccinia viral load. In addition, IL-17-deficient mice were more susceptible to the wild-type VV infection. In this light, the IL-23/IL-17-mediated antiviral immunity plays a subdominant anti-VV response.

IL-23 also helps upregulate the expression of several other immune effectors such as IL-21, IL-22, IFN-γ and the antiviral protein myxovirus resistance protein A (MxA) in peripheral blood mononuclear cells (PBMCs) in the course of virus infections (Figure 3). IL-21 was shown to be responsible for enhancing the expansion of CD8^+^ T memory stem cells (T_SCM_^S^) in inhibiting human immunodeficiency virus (HIV)-1 replication [28]. In chronic viral infections, IL-21 helps maintain the functions of CD8 T cells by inducing the expression of basic leucine zipper ATF-like transcription factor (BATF) through the STAT3-dependent pathway [29]. BATF, together with interferon regulatory factor-4 (IRF-4), up-regulates the expression of Blimp-1, which is crucial for the maintenance of virus-specific CD8 T cells. In order to further enhance antiviral activities in hosts, IL-23 limits the expression of transcriptional factors such as FoxP3 and GATA3, which are responsible for the differentiation of regulatory T cells (Treg). Eventually, the magnitude of the anti-inflammatory response that antagonizes IL-23-mediated pro-inflammatory reactions during virus removal is tightly regulated [30]. Collectively, IL-12 and IL-23 are crucial effector cytokines that promote a pro-inflammatory response for virus elimination in hosts.

Despite their remarkable antiviral properties, developing IL-12 and IL-23 into effective antiviral molecules requires greater understanding and more rigorous empirical evidence. This is especially important because both IL-12 and IL-23 mostly regulate antiviral activities via pro-inflammatory reactions, and it is proven that an immoderate inflammatory response is greatly associated with an increased risk of developing severe diseases or immune-directed complications [31]. Furthermore, owing to the superior adaptations to the host’s physiology, viruses have developed multiple antagonizing strategies to inhibit the synthesis or actions of effector cytokines in infected hosts. For instance, the binding of HIV-derived T20 peptide to the formyl peptide receptors on monocytes has been shown to inhibit the expression of IL-12 and thereby renders this anti-HIV defence ineffective [32].

Considering the importance of immunomodulation and the ever-evolving immune escape strategies of viruses, it is noteworthy to further explore the precise roles of IL-12 and IL-23 as part of antiviral therapies. For example, upregulating the upstream cytokine synthesis, which in turn enhances the expression of IL-12, could be an alternative to overcome the challenge. IFN-γ is responsible for initiating the transcription and translation of IL-12 in response to human herpes virus (HHV)-6 infections and the synthesis of IFN-γ can be tightly regulated via the inhibition of endogenous IL-10 [33]. Thus, manoeuvring cytokine synthesis involved in the positive or negative feedback loops might be able to facilitate antiviral activities in hosts. In addition, it has also been suggested that pro- and anti-inflammatory cytokines may be administered in pair, for instance, IL-12 and IL-35, in order to combat virus infections and control the magnitude of host immunity simultaneously [34]. While IL-12 functions to promote the host’s immunity against virus infections, IL-35 is co-administered to control the pro-inflammatory actions in order to prevent immune-directed complications in the host. Given the roles of IL-35 in regulating the host’s inflammatory response, its involvement in host antiviral immunity will be detailed in the next section. Of note, a precisely concerted immune response through pro- and anti-inflammatory cytokine actions can be one of the key antiviral approaches.

### 2.2. Anti-Inflammatory and Antiviral Actions of Interleukin (IL)-35

Considering the increasing number of emerging and re-emerging viral diseases, it is indispensable to embark on the effort to seek effective antiviral agents to control the spread of viral diseases [35,36,37]. As one of the newest members of the IL-12 family, the antiviral properties of IL-35 are actively being studied; nonetheless, there is very limited empirical evidence elucidating the practical use of IL-35 in treating viral diseases and their related complications compared to the other cytokine members.

IL-35 is composed of two subunits, namely Epstein–Barr virus-induced gene 3 (EBI3) and IL-12A (alpha subunit of IL-12) or p35 [38]. IL-35 shares the same alpha subunit with IL-12; therefore, it is able to exert similar inhibitory actions as IL-12 in combating virus infections. CD4+ T cells isolated from hepatitis B virus (HBV)-infected patients were shown to express higher levels of EBI3 and p35 mRNAs than the cells of healthy counterparts [39]. As aforementioned, IL-35 is able to exert anti-inflammatory effects to counterbalance pro-inflammation in hosts during virus infections. This prevents exacerbation of the viral diseases due to the vigorous immune response in the host. Under the influence of IL-35, anti-inflammatory Treg cells are formed to control and regulate overly reactive immune responses (Figure 4a). In hepatitis C, following the proliferation and activation of Treg, the synthesis of proinflammatory cytokines such as IFN-γ, IL-6, IL-8, and TNF-α was tightly regulated upon IL-35 treatment [40]. As a result, hepatitis C-associated liver inflammation was greatly improved, which manifested as a significant reduction in the serum ALT level and necrosis of hepatocytes [41].

Besides reducing immune-directed tissue damages and maintaining the physiological functions of vital organs, IL-35 also suppresses the differentiation of naïve CD8+ T cells into cytotoxic T cells (CTLs) via the downregulation of the costimulatory molecule CD28 (Figure 4b) [42]. In this process, the JAK1/TYK2/STAT1/STAT4 pathway is believed to concert the reaction cascade, while the reaction is governed by iTR35 cells [43]. It is thus said that the suppressive iTR35 is part of the regulatory elements that contribute to the downregulation of CTLs. Owing to the reduced number of mature CTLs, the intensity of cytotoxic and cytolytic events decreases, leading to less severe cellular damage [44]. In addition, iTR35 also suppresses the differentiation of Th2, particularly by regulating the production of Th2 cytokines [45].

Despite the exceptional immune-modulating properties of IL-35, deducing the pro- and anti-inflammatory mechanisms involved can be quite tricky. This is because any imbalance between pro- and anti-inflammatory immune responses can fail to eliminate the infecting virus; in the worst-case scenario, it possibly leads to undesired complications and disease manifestations [46]. Eventually, the infected person may undergo prolonged inflammation due to perpetual uncontrollable virus replication. This condition further results in chronic viral diseases and system failure [47]. Of note, in addition to its anti-inflammatory actions, IL-35 also induces antiviral activities via the synthesis of interferons which in turn undermines virus infections through several virucidal pathways. For instance, in influenza A, administration of commercially available recombinant human IL-35 (rhIL-35) was found to induce the production of antiviral cytokines such as IFN-α, β and λ1 (Figure 4c) [48]. Following the upregulation of antiviral interferons, the expression of antiviral proteins, including protein kinase (PKR), 2′,5′-oligoadenylate synthetase (OAS) and myxovirus resistance protein (Mx), is initiated. The activated PKR then phosphorylates eukaryotic initiation factor 2-alpha, (eIF2α) which is responsible for blocking viral protein synthesis [49]. OAS, on the other hand, is responsible for activating RNaseL endonuclease, which degrades viral RNA genomes [50]. To further enhance the antiviral activities in infected cells, MxA prevents the assembly of viral nucleocapsid protein (N), which is essential for viral genome transcription [51]. As a result, it inhibits the formation of new virus particles and stops the virus from spreading to neighbouring cells.

Indeed, IL-35 plays substantial roles in modulating the host immune response and eliminating infecting virus particles through various effector antiviral molecules. However, data pertaining to its exact mode of action are somewhat limited. In this light, more rigorous evidence-based research is required in order to better understand the mechanism of action of IL-35 in targeting virus infections and their disease complications so that IL-35 can be precisely applied as a prospective antiviral drug. Having the same EBI3 subunit as IL-35, a newly discovered but not fully studied member, IL-39, is worth noting for its antiviral potential. IL-39 is composed of p19/Ebi3 subunits and binds to the IL-23R/gp130 cellular receptor in order to orchestrate signalling pathways in affected cells. Activated B cells have been found to express IL-39 [52]. Additional empirical investigations are warranted to further disclose the role of IL-39 in the host immune response against virus infections.

### 2.3. Antiviral Properties of IL-27

IL-27 is a member of the IL-12 family and it consists of two subunits, p28 and EBI3 [53,54]. The compatible IL-27 receptor is also made up of two subunits, namely WSX-1 and gp130 (a shared receptor of IL-35). The receptor is found abundantly on the majority of cell types [55]. Immune cells such as B cells and macrophages are known to produce IL-27 [56,57]. IL-27 has shown promising antiviral potential due to its pro-inflammatory and anti-inflammatory properties. The duality of IL-27 can be explained as follows: IL-27-induced pro-inflammatory actions include upregulation of *CXC motif ligand-10 (CXCL-10)*, which is an IFN-γ-responsive gene, as well as increased expression of *transporter associated with antigen processing 1 (TAP1)* and *low molecular weight polypeptide 7 (LMP7)* genes that are actively involved in the antigen presenting process (Figure 4a) [58]. The interaction between CXCL-10 and its receptor CXCR-3 (also known as CXCL-10/CXCR-3 axis) promotes pro-inflammatory responses such as greater levels of pro-inflammatory cytokines and infiltration of inflammatory cells into the affected areas [59]. Meanwhile, the activation of the TAP1/LMP2/LMP7 pathway aids in processing and transporting intracellular peptides derived from viral antigens, which are subsequently associated with MHC class-I molecules to form complexes which are then displayed on the surface of antigen-presenting cells (Figure 5a) [60]. In addition, IL-27R-deficient NK cells were unable to restore their abilities to produce IFN-γ upon IL-27 stimulation [61]. It is highly possible that this phenomenon is due to the loss of intact receptors on the NK cell surface. Likewise, the level of IFN-γ was also reduced in mice whose *EBI3* gene was knocked out. These observations highlight two important key messages: (i) the structural intactness of IL-27 is mandatory to ensure cytokine signalling, and (ii) IL-27 is an important mediator for promoting the synthesis of pro-inflammatory cytokines such as IFN-γ and their downstream pro-inflammatory reactions.

In contrast, the anti-inflammatory properties of IL-27 are greatly associated with its ability to induce the proliferation of Treg cells and restore their suppressive capabilities [62]. As discussed earlier, Treg cells display anti-inflammatory effects via direct or indirect suppression on T cells in order to tightly regulate T-cell-mediated inflammatory responses (Figure 5b) [63]. IL-27 indeed has a broad spectrum of functions during a virus infection; whilst regulating the host immune response to prevent prolonged, damaging inflammation, it also functions to inhibit virus replication. In order to initiate antiviral activities in host cells, IL-27 activates the STAT-1/3 signalling pathway (Figure 5c). Upon the binding of IL-27 to the compatible cellular receptors, STAT-1 and -3 are phosphorylated and recruited in the cytosol of infected cells. The activated STAT-1 and -3 are then translocated into the cell nucleus to initiate the expression of antiviral genes, leading to inhibition of virus replication in infected cells [64]. This is well demonstrated through the inhibition of HIV replication by IL-27 via the STAT-1 signalling pathway, which promotes the synthesis of OAS-2 [65]. As mentioned earlier, OAS is responsible for activating RNase L, which digests viral RNA. Subsequently, the RNA genome of HIV is degraded, preventing the formation of new virus progeny. Likewise, the involvement of STAT-1/2/3 signalling has also been reported in IL-27-induced immunity against influenza A virus infection [66]. In the study, STAT-1, -2 and -3 were first activated through phosphorylation. The activation of STAT-1/2/3 then promoted the expression of PKR, which in turn inhibited the viral protein translation via eIF-2α. Besides OAS and PKR, the expression of MxA is also upregulated following the activation of STAT-1/3 signalling by IL-27 in hepatitis C [67]. Ultimately, the inhibitory actions circumvent the production of new virions and the spread of the virus infection in the host.

In addition, IL-27 also promotes the expression of chemokines such as IFN-γ-inducible protein-10 (IP-10) and monokine-induced-by-interferon-γ (MIG) through the activation of the STAT 1/3 signalling pathway (Figure 4c) [68]. IP-10 and MIG attract T lymphocytes, especially cytotoxic T lymphocytes, to the affected area in order to remove infected cells and prevent the spreading of the virus infection [69]. Interestingly, the antiviral activities of IL-27 are not only restricted to human viruses; it was also shown to inhibit fowl plague virus infection in birds. When targeting the fowl plaque virus infection, STAT-1 signalling was initiated by IL-27 in order to elevate the expression of antiviral products to reduce the viral load in infected birds [70]. More recently, a study on Zika virus infection showed that IL-27 induced the expression of antiviral genes in vitro and in vivo in a STAT-1-dependent manner [71]. These studies strongly suggest that IL-27 is able to execute immune defence against a broad range of virus infections, especially in STAT-dependent manners, in order to combat virus infections in vitro and in vivo.

## 3. Future Prospects of Cytokine Therapy in Combating Virus Infections

Both the immune-modulating and antiviral properties of cytokines point to the vast potential of cytokine therapy in treating virus infections. For instance, interferons are able to inhibit virus replication via multiple innate immune responses. It was clearly demonstrated that treating moderate COVID-19 cases with IFN-α2b could accelerate viral clearance while maintaining adequate levels of circulating pro-inflammatory cytokines such as IL-6 and CRP (C-reactive protein) to strengthen the patients’ immunity against the disease [72]. In addition, the combination of ribavirin and IFN-α in chronic HBV treatment was reported to induce the proliferation of CD4+ T cells that produce various signalling cytokines to initiate antiviral actions in infected cells and to alert neighbouring cells of the threatening virus infection [73]. Patients infected by parvovirus B-19 treated with IFN-β also showed a significant reduction in viral load in the follow-up endomyocardial biopsies (EMBs). The patients continued to show improvement in their hemodynamic profiles six months post-treatment [74]. Recently, a randomized controlled trial of IFNβ-1a and IFNβ-1b among severe COVID-19 patients showed encouraging findings, particularly the lower in-hospital mortality rate among the admitted patients. The Time-To-Clinical-Improvement (TTCI) was also significantly improved compared to those in the control group [75].

In view of the excellent treatment outcomes of interferon-based antiviral therapies, it has been suggested that interferons may be effective mediators in IL-12 family-based cytokine therapy for treating virus infections in order to ensure promising therapeutic outcomes. In addition to the aforementioned interferon-α and -β, IFN-γ also has a unique ability in promoting antiviral immune responses [76]. Interestingly, IFN-γ is associated with the increased expression of IL-35 in human tolerogenic DCs (tolDCs) [77]. tolDCs direct immunotolerance in hosts in order to better modulate pro-inflammatory reactions. During immunotolerance, IL-35 induces the proliferation of iTr35 and Breg cells that primarily control immune-mediated actions [78,79]. As such, iTr35 helps restore immune homeostasis and prevent immune-mediated complications [80]. The suppressive function of IL-35 was evidently demonstrated in chronic hepatitis B in which the differentiation of pro-inflammatory Th17 was restricted, hence reducing inflammation in the liver [81]. Considering the promising immune-modulating properties of IFN-γ and IL-35, the combination of these cytokines in antiviral therapies could be a new alternative to existing treatments. Combination of IFN-γ and IL-35 was attempted in a murine model [82] that investigated the allogeneic reaction of graft-vs-host (GvHD) disease. IFN-γ and IL-35 were applied ex vivo to the graft prior to the transplantation. The treatment then successfully reduced the severity of acute graft-vs-host disease (aGvHD) through the expansion of Treg and diminished Th1 functions. A similar concept is applied when utilizing IFN-γ and IL-35 together in antiviral treatments, in which the former directs antiviral immunity whereas the latter simultaneously modulates the anti-inflammatory response to prevent a damaging inflammatory reaction; as a result, an effective yet less harsh antiviral approach can be achieved.

While many antiviral benefits can be achieved through cytokine therapy, many aspects, such as frequency, timing, and dosing of therapy, need to be accurately examined. This is because overstimulation or understimulation of the immune response due to cytokine administration may possibly result in unexpected side effects which may worsen the patient’s conditions. For example, the toxicity of IL-12 is highly dependent on the schedule of cytokine administration [83]. In addition, the side effects of IFN-based therapy in hepatitis C patients, such as the occurrence of pneumonitis, have also been described elsewhere [84]. In many circumstances, monoclonal antibodies such as ustekinumab, secukinumab and ixekizumab are used to treat cytokine-induced symptoms and side effects. When considering cytokine therapy for virus infections, it is noteworthy to include the aforementioned health-concerning issues in order to maximize the therapy effects without compromising the patient’s health. Hence, detailed studies are needed to develop a safe and effective cytokine treatment in inhibiting virus infections and reducing disease severity.

## 4. Conclusions

Viruses are pathogens that can infect humans and animals through a variety of routes. Some of them can be infectious and pathogenic enough to cause serious illnesses in infected hosts. Besides direct cytocidal effects, virus infections also induce severe and chronic inflammation that adds to the severity of diseases. The more prolonged the inflammation, the more severe the complications the host may encounter. In addition, on many occasions, viruses are able to develop immune-escape mechanisms to evade the host’s immune surveillance and this further complicates the disease treatment and management; worse comes to worst, it can result in antiviral resistance in viruses. In spite of the growing number of emerging and re-emerging viral diseases worldwide, effective antivirals available for treating virus infections are still very limited. Cytokines belonging to the IL-12 family are capable of triggering the production of effector molecules such as chemokines, interferons and antiviral peptides to attenuate virus infections. Some members such as IL-27 and IL-35 even exhibit dual effects, i.e., pro- and anti-inflammatory responses in the course of virus infections. Despite the promising findings, many biological functions of cytokines are yet to be discovered, including their involvement in the host’s antiviral defence. As a result, it is important to determine the precise mechanisms of actions of cytokines in targeting virus infections, as neither prolonged anti-inflammatory responses nor chronic pro-inflammatory responses will benefit infected persons. Moreover, an imbalanced immune response may cause irreversible tissue damage and organ failure, as well as delayed virus clearance. Conclusively, developing cytokines into effective antiviral therapeutics is a challenging journey and requires rigorous investigations in order to ensure safe, long-lasting and effective antiviral formulations for viral diseases.

## Figures and Tables

**Figure 1 ijms-24-07350-f001:**
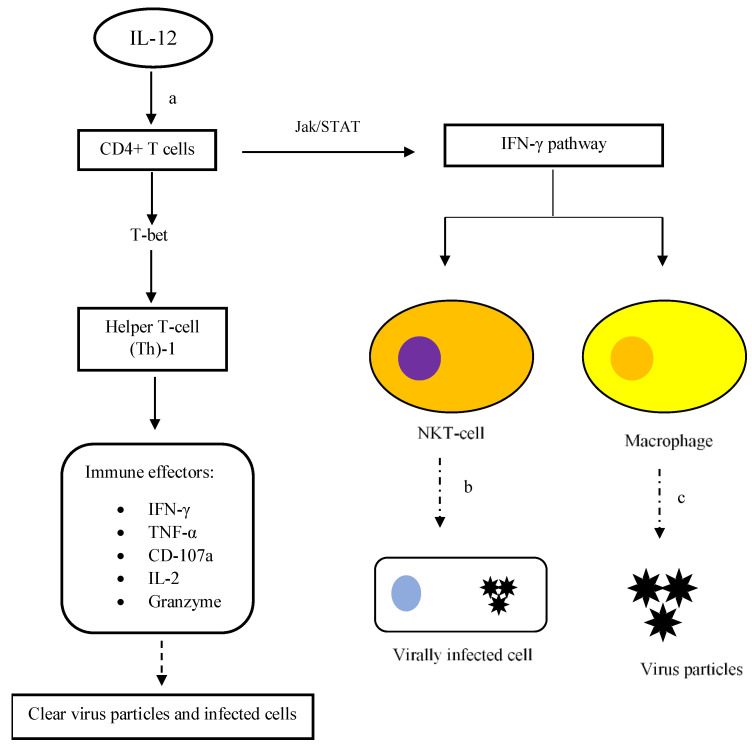
Activation of CD4+ T cells and IFN-γ pathway via IL-12 signalling. (**a**) Upon binding to cell receptors, IL-12 initiates the proliferation of CD4+ T cells into Th-1 cells that subsequently activates immune effectors to further enhance host defence against virus infections. Eventually, clearance of virus particles and infected cells can be carried out by the host immune system. Secondly, the activated Th-1 cells also synthesize IFN-γ via Jak/STAT signalling. The release of IFN-γ into the surroundings encourages the proliferation and activation of macrophages and NK T cells. (**b**) NK T cells are important cellular effectors in eliminating infected cells. (**c**) Once armed, macrophages actively engulf virus particles to prevent virus spread. The dashed-line arrows represent the antiviral activities.

**Figure 2 ijms-24-07350-f002:**
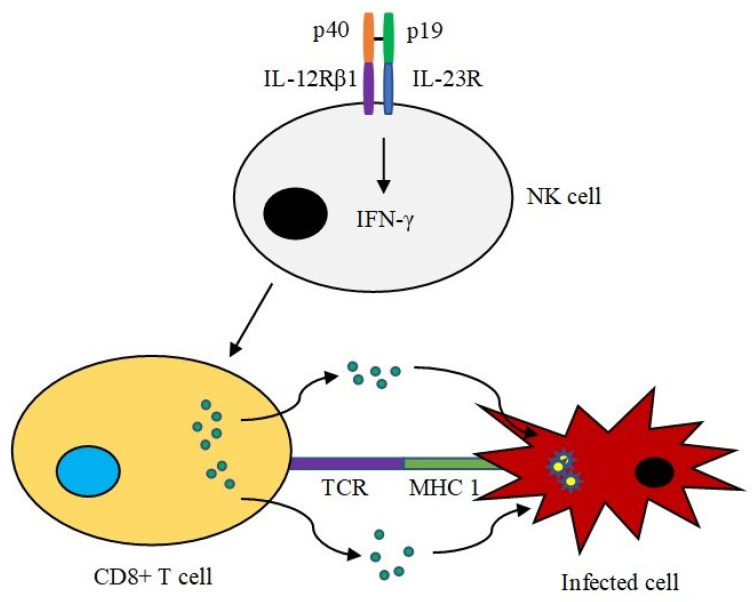
IL-23 activity in NK cells. IL-23 (p40/p19) binds to its receptor on the surface of NK cells. The binding leads to the production of effector IFN-γ. IFN-γ is responsible for promoting CD8+ T-cell responses such as eliminating infected cells in a perforin-dependent manner. The targeted cell undergoes programmed cell death, preventing spread of the virus.

**Figure 3 ijms-24-07350-f003:**
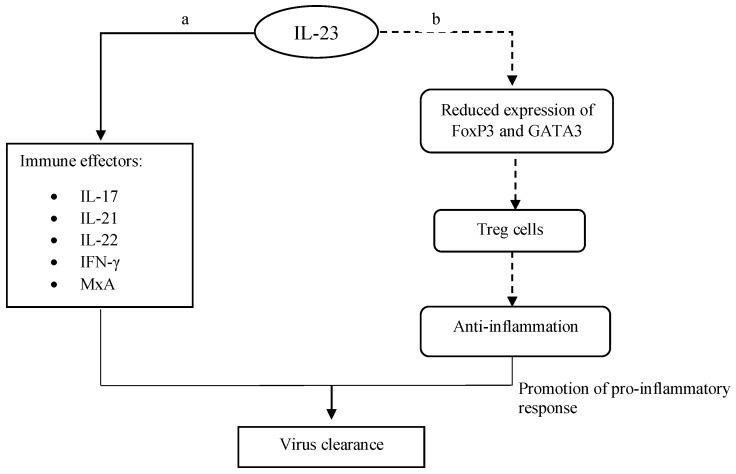
Pro-inflammatory reactions by IL-23. (**a**) IL-23 induces the expression of immune effectors. Upon their expressions, those effectors enhance the host immune response, which subsequently elevates virus clearance. (**b**) IL-23 limits anti-inflammatory activities (dashed-line arrows) via the downregulation of Treg. It later increases the pro-inflammatory response that aids in the removal of virus particles.

**Figure 4 ijms-24-07350-f004:**
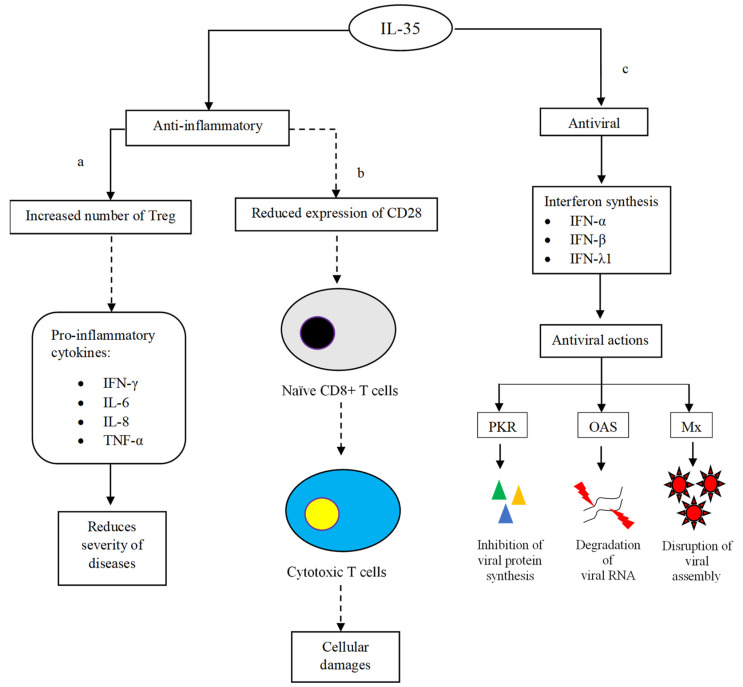
Anti-inflammatory and antiviral actions of IL-35. (**a**) IL-35 induces the proliferation of Treg in order to suppress the synthesis of pro-inflammatory cytokines. The inhibition subsequently improves inflammation and tissue damage that are usually manifested as the severity of virus infections. (**b**) IL-35 controls the maturation of naïve CD8+ T cells into cytotoxic T cells. The limiting number of mature cytotoxic T cells prevents the occurrence of cellular damage. (**c**) IL-35 induces antiviral effects via interferon production. Interferons are known to inhibit viral protein synthesis, genome replication and assembly. The inhibitory activities prevent virus replication and limit virus spreading. The dashed-line arrows represent inhibitory effects.

**Figure 5 ijms-24-07350-f005:**
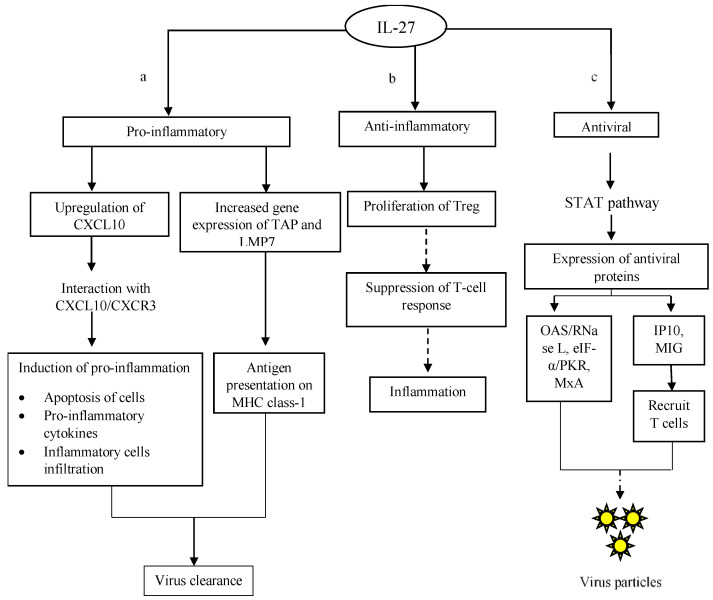
IL-27-mediated antiviral response in infected cells. (**a**) IL-27 triggers the production of effector mediators and chemokines to promote inflammation. These mediators enhance the host immune response in its targeting of virus infections, for example antigen presentation, cell apoptosis, synthesis of pro-inflammatory cytokines and infiltration of immune cells. Those pro-inflammatory reactions are responsible for virus clearance. (**b**) In order to control the occurrence of pro-inflammations in infected cells, IL-27 promotes the formation of Treg cells that execute anti-inflammatory responses in the host. Treg controls T-cell-mediated inflammation in order to prevent undesired immune-mediated complications. (**c**) Upon binding to receptors, IL-27 activates STAT signalling pathways. The activation of STAT signalling results in the transcription and translation of antiviral products that help target virus replication. The dashed-line arrows indicate inhibitory actions on cellular inflammation and the synthesis of new virus progeny.

## Data Availability

Not applicable.

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
