# Peer review of "Potential of Interleukin (IL)-12 Group as Antivirals: Severe Viral Disease Prevention and Management"

_ijms, 2023, doi:10.3390/ijms24087350_

Round 1

Reviewer 1 Report

The authors are reviewing the IL-12 group of cytokines, aiming to focus on their anti-viral activities.

To begin with, I would recommend the inclusion of IL-39, even though it is not yet fully studied. I would also recommend a more systematic, rather standardized approach of all these cytokines that should include structural details of both the cytokines and their respective receptors, as well as the signaling pathways, what cells are secreting the respective cytokines and under which circumstances, which cells are expressing the corresponding receptors.

Line 37. Even though the cited article mentions the pro-inflammatory effect of IL-10, I would tend to disagree as many more references state the inhibiting role of this cytokine.

Line 39. The authors mention the particular effects of SARS-CoV-2 upon some cell types. However, the context was in fact larger, discussing viral infection in general. Should the authors decide to give such details and examples, they should perhaps consider giving details about many more viruses, especially since such viruses are mentioned later in the manuscript and SARS-CoV-2 is not.

Line 65. These instead of those

Line 74. Since we are talking about a review, this representation can be considered rather schematic and I encourage the authors to give several more details regarding the sensor cells, the involvement of PRR (PAMP recognition receptors), the release of IL-12 by particular cells etc. The text, as it is, suggests a T cell activation regardless of the TCR specificity, but dependent of the TCCR (T cell cytokine receptors).

Line 85. I would suggest rephrasing. For instance, Bhardwaj et al. showed …

Line 114. Delete also

Line 116. Replace was with is

Discussing the interplay between IL-17 and IL-23, the authors decide to exemplify and to mention the experiments with vvIL-23. However, I feel that this subject needs to be expanded, given the available data regarding the fact that Th17 cells require IL-23 mediated RORγT stabilization in order to proliferate and to maintain their activity. From this point of view, IL-23 antagonizes the IL-12 effects, even though it is a member of the IL-12 family and the IL-23 receptor is a heterodimer sharing a chain with the IL-12 receptor. Furthermore, one should then consider the cells able to produce large amounts of IL-23 (monocytes/macrophages, dendritic cells, innate lymphoid cells, but also B cells and γδT cells), and under what circumstances (the IBD gut for example). Discussing inflammation, one should bear in mind that IL-17 limits the Th1 development, while Tc17 cells are not considered as being involved in viral clearance (or at least they are not critical for this process). Furthermore, IL-17 can be mentioned in the context of immune evasion mechanisms being able to increase virulence. I would also mention, in the context of anti-viral defense, the expression of IL-23R on NK cells.

Line 117. You mention the up regulation of several soluble factor. I did not understand the context. Is it related strictly to the vvIL-23 experiments? Which cells are involved in secreting these cytokines, especially since you mention IFNγ?

Line 126. You mention FoxP3 and GATA3 as “anti-inflammatory factors”. These are transcriptional factors.

Line 155, 160. Why is his discussed here and not after IL-35 is presented as well?

Line 177. The authors state that IL-35 exerts similar immune regulatory functions as IL-12. To the best of my knowledge, while IL-12 is secreted by pro-inflammatory effects, IL-35 is rather secreted by Tregs and Bregs (leaving aside some other cell types such smooth muscle cells and endothelial cells), leading to the stimulation of these regulatory cells and the down activation of Th17 cells.

Line 179. Please explain p53 and its role in this context

Line 187. Please rephrase “… which were signified …”

Line 193. Please offer details about iTR35 cells, in the sense that they are IL-35 producing regulatory cells targeting mostly Th2

Line 199. Which two arms of the immune response are addressed here: Innate/acquired or cellular/humoral?

Line 204. Please mention under what circumstances occurs the production of IL-35. Please mention also that the IL-35 administered is a commercially available products, rhIL-35.

Line 234. Which cells are producing IL-27?

Line236. You may consider worthwhile mentioning that gp130 is used by IL-35 as well.

Line 237. This cytokine plays indeed a dual role. However, this duality does not explain in itself the antiviral role.

Lines 239 and 245. I would suggest you put together the information regarding the IFNγ and β.

Line 244. I tend to disagree that apoptosis is part of inflammation per se. By extension, anti-inflammatory drugs would be abe to prevent apoptosis.

Line 254. In this conext, can you make a comparison with the other receptors you are discussing in your manuscript?

Line 259. Ths is controversial. Please see Meka et al. 2015

Line 269, 270. I would suggest pooling these 2 sentences.

Reviews regarding this group of cytokines are already available, but this manuscript, according to the title, was pushing things further, reaching to potential treatment based on these cytokines. Hence, the major objection I have, since this manuscript is aiming to the description of „viral disease prevention and management” is that the chapter dedicated to this specific aim is not developed enough, in my opinion. Furthermore, the authors should dedicate dedicate more space to the complex interactions and multiple effects of these cytoines on various types of cells (pleiotropism). One should approach as well the hepatitis IFN-based and the IFN-free treatment as a model of how side-effect might hinder the efficiency of a cytokine-based treatment. There are some very interesting data about monoclonals directed to these cytokines (ustekinumab, secukinumab, ixekizumab, brodalumab, guselkumab, tildrakizumab, risankizumab) Etc.

Author Response

Dear respected reviewer,

Reviewer 2 Report

The review wrote “Potential of Interleukin (IL)-12 Group As Antivirals: Severe Viral Disease Prevention and Management. The review is interesting but has several major issues that must be addressed. Here are my comments:

1.      Abstract should be revised as it has several abbreviations, which are difficult for the general reader to understand the aims of the review. In the abstract section, the authors should write the background of the review. What motivated the researcher to do this review? Why is this review important?

2.      Introduction is very short; authors need to elaborate on each section, and authors should provide additional references.

3.      Authors should write that cells produce IL-12 and include appropriate references.

4.     Several other effects (i.e., IL-12) could be related to other inflammatory disorders.

5.     Authors should write a paragraph on natural T regulations and show them diagrammatically.

6.     Explain the role of IL-12 in experimental animal models.

7.     This review is too speculative; more research studies should be described to validate the conclusions. I suggest the authors revise the conclusion portion.

8.     Authors should add the latest references.

Author Response

Dear respected reviewer,

Round 2

Reviewer 1 Report

Thank you for addressing the issues raised during revision.

I would suggest a minor change in the abstract:  ...meanwhile preventing

As I mentioned before, I also consider that the manuscript would have benefited from a more systematic, almost standrdized approach of the cytokines and ctokine receptor structures and their signalling pathways. Nonetheless, the manuscript was significantly improved and I can endorse its publication.

Author Response

Dear reviewer:

Reviewer 2 Report

The authors have answered my questions, and the paper has significantly improved.

Author Response

Dear reviewer,
